# Learning Bimanual Scooping Policies
# for Food Acquisition

**Jennifer Grannen\*, Yilin Wu\*, Suneel Belkhale, Dorsa Sadigh**
Stanford University, Stanford, CA
{jgrannen, yilinwu}@stanford.edu

**Abstract:** A robotic feeding system must be able to acquire a variety of foods. Prior bite acquisition works consider single-arm spoon scooping or fork skewering, which do not generalize to foods with complex geometries and deformabilities. For example, when acquiring a group of peas, skewering could smoosh the peas while scooping without a barrier could result in chasing the peas on the plate. In order to acquire foods with such diverse properties, we propose *stabilizing* food items during scooping using a second arm, for example, by pushing peas against the spoon with a flat surface to prevent dispersion. The added stabilizing arm can lead to new challenges. Critically, this arm should stabilize the food scene without interfering with the acquisition motion, which is especially difficult for easily breakable high-risk food items like tofu. These high-risk foods can break between the pusher and spoon during scooping, which can lead to food waste falling out of the spoon. We propose a general *bimanual scooping primitive* and an *adaptive stabilization strategy* that enables successful acquisition of a diverse set of food geometries and physical properties. Our approach, CARBS: Coordinated Acquisition with Reactive Bimanual Scooping, learns to stabilize without impeding task progress by identifying high-risk foods and robustly scooping them using closed-loop visual feedback. We find that CARBS is able to generalize across food shape, size, and deformability and is additionally able to manipulate multiple food items simultaneously. CARBS achieves 87.0% success on scooping rigid foods, which is 25.8% more successful than a single-arm baseline, and reduces food breakage by 16.2% compared to an analytical baseline. Videos can be found on our website.

**Keywords:** Bimanual Manipulation, Food Acquisition, Robot-Assisted Feeding, Deformable Object Manipulation

## 1 Introduction

Approximately one million people in the U.S. depend on a caregiver's assistance to eat [3], which can lead to malnutrition [4, 27] and an erosion of self-worth [23]. Building a robotic feeding system would enable patients to eat food independently [25]. A key component of such an assistive feeding system is bite acquisition, i.e., the act of a robotic arm picking up morsels of food from a plate for the goal of transferring the food to person's mouth [8]. Prior strategies for bite acquisition acquire food using a single robotic arm for either skewering with a fork or scooping with a spoon. Fork-based bite acquisition learns to select skewering primitive parameters from a large supervised dataset of food items [1, 6, 11]. However, fork-based skewering is inherently limited in what foods it can acquire. For example, a fork cannot skewer brittle cashews or small peas without damaging the food item, while scooping with a spoon might be more successful. Existing single-arm spoon-scooping bite acquisition often uses a hard-coded single-arm scooping primitive, making generalization to varied food items difficult [21, 22]. When scooping more diverse foods, such as large, rigid-body fruit cubes and deformable cottage cheese, prior works rely on hard-coded adaptation strategies – new primitives and even new tools – which are not scalable. Similar to fork skewering, the single-arm scooping strategy is also inherently limited. For unstable items like broccoli or blueberries, it can be hard to know where exactly to scoop without pushing it off a barrier such as a fork or bowl wall.

---

*Equal contribution.

6th Conference on Robot Learning (CoRL 2022), Auckland, New Zealand.

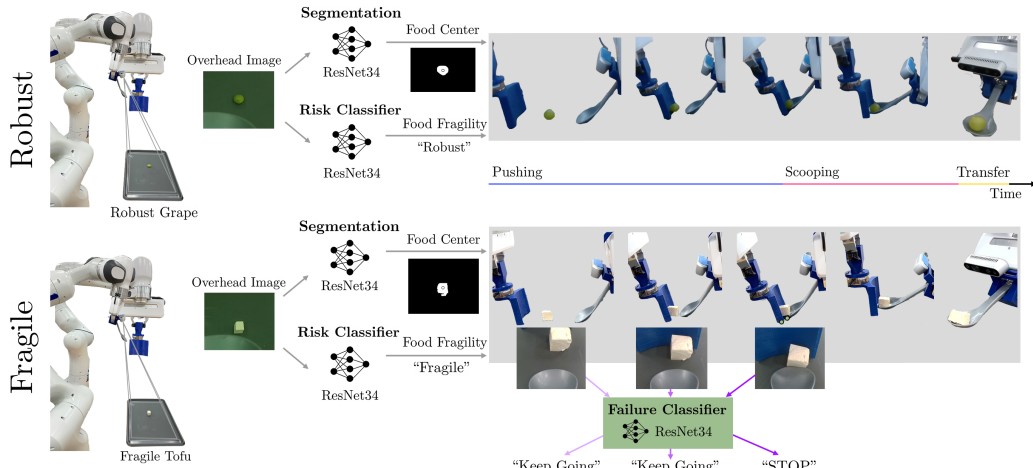

Figure 1: **Learned Bimanual Scooping**: CARBS is a bimanual scooping system for foods with varied geometries and deformabilities. CARBS uses a second arm to stabilize food position during scooping. To avoid breaking deformable foods between the two arms, we learn a Risk Classifier and a Failure Classifier to identify high-risk, fragile foods and breakage-imminent states respectively.

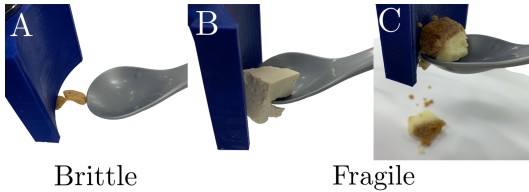

Figure 2: **Bimanual Failures**: Adding a second stabilizing arm during scooping can lead to new breakage failures for foods of varying deformabilities. For both brittle and fragile food items like cashews (A), tofu (B) and cheesecake (C), food pieces can become wedged between the two arms and break due to excessive force being applied between the pusher and scooper. These breakages can cause food to fall out of the scooper (C), leading to food waste and scooping inefficiencies.

Scooping objects with unstable geometries or scooping multiple objects without the help of a barrier – for example, chasing peas on a plate – is nontrivial and inefficient. To develop a generalizable scooping policy that can work across foods with difficult geometries, we need to use an additional robot arm to *stabilize* the food item. For example, a primary arm should scoop peas against a surface that is pushing towards the spoon to stabilize the pea positions (See Fig. 1). As humans we intuitively and regularly use two arms to *stabilize the environment* in many everyday tasks: when tying our shoes, we use a second arm to hold the knot; or similarly when cutting a steak we use a second arm to hold the steak to make the act of cutting easier for the first arm. As a result, there has been a growing focus on bimanual manipulation for tasks including rope untangling, peg insertion, food cutting, fabric manipulation, bottle opening, and bag opening, all implicitly leveraging the same insight that an additional arm can act in a stabilizing manner [5, 9, 12, 24, 28, 29]. In bite acquisition, we posit a second arm holding a *pusher* (as shown in Fig. 1) can support a scooper in scooping objects with difficult geometries and deformabilities.

However, adding a second arm for stabilizing the environment – holding or pushing the food items – opens the door to a new set of complications and failures (see Fig. 2): A pushing arm needs to physically make contact with the food to stabilize it. This can easily break or deform food items during scooping and thus impede on task progress. For objects with unstable geometries like snowpeas or macaroni, it is helpful for the barrier to follow the food into the scooper so the food does not fall out as the scooper finishes scooping, but this design comes at a cost. Fragile and deformable items like tofu or jello cubes can easily break when forced into the scooper by the pusher, which can also cause the item to fall off the scooper or leave residue on the plate. As a result, it is difficult to define a single hard-coded primitive to that generalizes to both unstable geometry and breakage failures. We posit that many breakage-prone, or "high-risk" foods will break under predictable scenarios when the pusher and scooper are squeezing the deformable food together. We employ this idea when scooping high-risk foods by detecting "breakage-imminent" states and adjusting our scooping policy to anticipate and prevent food breakage and waste.

*Our key insight is we need a second arm to effectively stabilize food environments by identifying high-risk food items, i.e. the ones that are breakable and fragile, and adapting a dynamic stabilizing strategy to anticipate and prevent such failures.*

In this work, we propose a learned bimanual scooping policy, CARBS: Coordinated Acquisition with Reactive Bimanual Scooping, that uses a scooper arm and a pusher arm (similar to fork or knife) to acquire food items of a wide variety of properties, including shapes, sizes, and deformability. CARBS first identifies *high-risk food items* by classifying a visual observation of the target food item as fragile or robust. Next, CARBS prevents breakage and dropping failures by *servoing to adapt* the stabilizing action parameter: the distance between the scooper and pusher. This paper makes the following contributions:

**A Bimanual Scooping Primitive for Bite Acquisition.** To our knowledge, we are the first work to study bimanual strategies for acquiring widely varied foods. We define a novel bimanual scooping primitive and show that it generalizes to scooping 14 food classes with varied geometries and deformabilities. Our primitive is also the first to scoop multiple food items per acquisition action.

**Learning to Avoid Bimanual Failures.** We contribute a framework for anticipating and preventing bimanual failures by identifying high-risk scenarios and adjusting the stabilizing parameter accordingly. To prevent breakage failures during bimanual scooping, this entails identifying deformable, fragile foods and adjusting the distance between arms during scooping. When scooping high-risk foods such as cheesecake and tofu, we learn to detect hazardous, breakage-imminent scooping states for closed-loop visual feedback to adjust our dynamic stabilizing policy.

**Evaluating Learned Scooping Policies.** We present physical experiments with CARBS, which learns to identify fragile foods and adapt a stabilizing parameter to execute a bimanual scooping action. We find CARBS is able to successfully generalize across food geometries and deformabilities to scoop 14 food classes without breakage in 85.7% of trials. We will also open-source our food fragility and breakage datasets and the pusher and scooper CAD models.

## 2  Related Work

**Food Acquisition.** Previous works have studied single-arm food acquisition with chopsticks [17, 18], skewering with a fork [1, 6, 8, 11], and scooping with a spoon [21, 22]. Ke *et al.* [18] study grasping a set of household objects with chopsticks, but do not consider food objects or variations in geometric and physical properties. Past works on fork-based bite acquisition take a step towards more general food acquisition by learning an optimal skewering policy from a large dataset of food items [1, 6, 11]. However, both chopsticks and forks with their accompanying acquisition primitives may struggle to generalize to many food items. In particular, it would be difficult to acquire fragile or very small foods such as jello or peas with a fork or chopsticks because the foods could break or would require very precise acquisition strategies unforgiving of slight errors. Spoon scooping is a promising acquisition alternative for these foods. Ohshima *et al.* [21] consider scooping portions off a large block of deformable foods – tofu and pudding – with an analytical policy. However, this analytical single-arm scooping method is limited to food items of relatively homogeneous geometries and deformability. Park *et al.* [22] also use a single arm to scoop more varied foods (i.e. fruit cubes, cottage cheese) from a bowl by asking a human end-user to select between three spoon tools of differing materials and shapes per food class, which is not scalable to the large universe of potential foods. In contrast, our learned bimanual scooping approach uses one scooper and one pusher tool to acquire foods of highly variable geometries and deformabilities off a plate.

**Bimanual Manipulation.** In recent years, bimanual manipulation has enabled robots to perform new tasks using new motion primitives such as bag opening, flinging fabrics, cutting vegetables, and opening bottles [2, 5, 9, 12, 13, 16, 20, 29]. The extra mobility provided by an additional arm can take the role of stabilizing the environment to reduce non-stationarity and make the task easier. For example, past works have implicitly considered utilizing a stabilizing arm for tasks including peg insertion, untangling ropes, and cutting foods [5, 9, 12, 24, 29]. This idea is very relevant to the food manipulation domain. Foods can have widely varied physical properties and geometries – they can be deformable, brittle, slippery, and in unstable shapes and poses. As a result, it can be difficult to manipulate these objects due to the unpredictable dynamics of food movements and interactions.

Past bimanual food manipulation works have considered food preparation tasks – cutting and peeling vegetables, scooping out a melon, and mixing in a bowl. Food cutting or peeling works use a stabilizing arm to hold the food in place during the cutting motions, but these stabilizing strategies are largely stationary or analytical [9, 29] and do not make additional task progress [7]. On the other hand, Ureche *et al.* [26] use a more versatile stabilizing strategy for melon scooping, zucchini peeling, and bowl mixing by learning bimanual interaction constraints from human demonstrations.

Figure 3: **Bimanual Stabilizing Strategies**: CARBS uses 3 stabilizing strategies: (1) *Angled Pushing* and (2) *Adaptive Cupping* during the Pushing Phase, and (3) *Pinning* during the Scooping Phase. In *Angled Pushing*, the pusher moves at an angle $\theta = 15°$ off the vertical, which encourages food items to roll off the barrier surface and into the spoon. In *Adaptive Cupping*, the pusher pushes foods towards the spoon with the concave surface and cups them to be centered with the spoon mouth. This strategy is parameterized by a learned parameter $\alpha$ that represents the scaling of the distance travelled by the pusher. In *Pinning*, the pusher moves upwards to follow the spoon mouth as it scoops to prevent food items from toppling and falling out.

For melon scooping, the stabilizing arm holding the melon is able to adjust its force to brace against the scooping tool to better stabilize the melon's position and even progress towards task success by pushing more melon into the scooper. However, this work is only specialized for scooping a melon and does not consider a generalizable policy for the large variety of food items necessary for a scalable robotic feeding system. Our method similarly uses a dynamic stabilizing strategy that pushes food towards the scooper to both stabilize food position and make more task progress. However, our method is able to learn to directly sense states close to bimanual constraint violations (food breakage failures) from visual feedback, and bypasses the need for expensive human demonstration collection. Our method is also able to learn these constraints in a food-agnostic way and generalizes to visually varied, out-of-distribution food classes.

## 3 CARBS: Coordinated Acquisition with Reactive Bimanual Scooping

We consider the task of scooping a variety of food items off a plate while maximizing the integrity of the food item, i.e., the weight, after the scooping motion. Food items may vary in geometry and deformabilities, including foods that are brittle (i.e. cashews), compliant (i.e. pasta), and fragile/breakable (i.e. tofu). We assume access to a plate workspace with a standard $(x, y, z)$ coordinate frame (as in Fig. 1). We assume full bimanual access to this plate workspace with the following two mounted tools: *Scooper* and *Pusher* (See Fig. 4). The *Scooper* tool is a plastic spoon mounted at an angle to the robot end effector with a camera mounted for access to angled images $I \in \mathbb{R}^{W \times H \times C}$ of the spoon and surrounding workspace. The *Pusher* tool is a concave barrier that is mounted vertically to the robot end effector[1].

We model the bimanual scooping task as a Partially Observable Markov Decision Process (POMDP) $\mathcal{M} = (\mathcal{O}, \mathcal{S}, \mathcal{A}, \mathcal{T}, \mathcal{R})$. We observe images $I \in \mathcal{O}$ of the unknown food environment states $S \in \mathcal{S}$ and define an action space $\mathcal{A}$ as joint 14-DoF robot actions $(a^s, a^p)$. The visual state space is an RGB image observation space $O \in \mathbb{R}^{W \times H \times C}$. We assume unknown transition dynamics $\mathcal{T} : \mathcal{S} \times \mathcal{A} \to \mathcal{S}$, an initial state distribution of food configurations $\rho_0$, and a time horizon $T$. We define a reward $\mathcal{R} : \mathcal{S} \times \mathcal{A} \to \mathbb{R}$ as the weight of the food in the scooper after scooping. We aim to construct a closed loop policy $\pi : \mathcal{O} \to \mathcal{A}$ to maximize the expected reward by successfully scooping a set of varied foods.

To scoop a large diversity of foods with two arms without dropping or breaking them, we introduce CARBS, a reactive bimanual scooping policy learned from real food interactions (Fig. 1). We simplify the complex bimanual action space by introducing a novel bimanual primitive, which is parameterized by the *distance of travel for the pusher* and the location of the food item, and employs three bimanual stabilizing strategies: (1) *Angled Pusher*, (2) *Cupping Motion*, and (3) *Pinning Motion* (shown in Fig.3). We show that this parameterization generalizes to robust food items (e.g., grape) by selecting a large distance of travel, as well as for breakage-prone items (e.g., tofu) which require more adaptive pusher travel distances. To handle the latter case of breakage-prone items,

---

[1]This design is inspired by antique pushers that were used to push foods into spoons.

CARBS learns to adapt the pusher travel distance by anticipating common types of failure. Our policy network learns to identify which regime we are in (robust or breakage-prone) based on just the initial plate image.

**A Bimanual Scooping Primitive.** We define a parameterized bimanual scooping primitive that takes two inputs, pusher travel distance $\alpha$ and food position $(x_f, y_f)$, and has three phases to be performed in succession: (1) *Pushing*, (2) *Scooping*, and (3) *Food Transfer*. We thus reduce the 14 DoF action space $\mathcal{A}$ to 3 dimensions: pusher travel distance $\alpha$ and food position $(x_f, y_f)$. The scooper and pusher begin in starting positions centered around the food position along a fixed pushing axis. We empirically found that the choice of pushing axis did not affect performance, so we select the $x$-axis to favor our robots' range.

During the *Pushing* phase, both the pusher and scooper move towards each other along the $x$-axis towards some point $p_{\text{push}} = (x_f + d, y_f)$ closer to the scooper than pusher, as shown in Figure 3.A-B. The Pushing phase also utilizes two bimanual stabilization strategies: *Angled Pusher* and *Cupping Motion* (See Fig. 3.A-B). In Angled Pusher, the pusher is angled at some fixed angle $\theta = 15°$ about the $y$-axis, rather than orthogonal to the plate. This stabilizes the food position by encouraging the food to slide or roll into the scooper at the end of pushing when the two arms meet, as shown in Fig. 3.A. In Cupping Motion, the pusher pins the food against the concave surface as it moves towards the scooper, which promotes centering the food position during movement into the entrance of the spoon (see Fig. 3.B). Cupping Motion is an adaptive stabilizing strategy that takes as input the primitive input $\alpha \in [0, 1]$, which determines how close the pusher and scooper get to each other. $\alpha = 0$ implies no motion and $\alpha = 1$ results in the pusher and scooper reaching each other.

After the Pushing phase, we assume CARBS has successfully manipulated the food item into the scooper and move to the *Scooping* phase. In this phase, we rotate the scooper up about the $y$ axis to "scoop" the food into the bowl of the spoon. During the Scooping phase, CARBS employs the *Pinning Motion* stabilizing strategy where the pusher moves up along the $z$ axis with the scooper as it rotates (shown in Fig. 3.C). This strategy prevents foods in unstable poses within the spoon from falling out of the scooper. Lastly, CARBS finishes with the *Food Transfer* phase by moving the pusher away from the scooper and rotating the scooper towards an end user to prepare for feeding.

**Identifying High-Risk Settings.** To learn the inputs of the bimanual scooping primitive $(x_f, y_f)$ and $\alpha$ from image observations $I \in \mathcal{O}$, CARBS leverages the insight that foods with similar deformabilities encounter similar breakage failures, and posits that learning to identify high-risk settings, e.g. robust vs. breakage-prone foods, will help determine the optimal $\alpha$ inputs. Due to the high variability of food dynamics, we do not learn a dynamics model of our POMDP $\mathcal{M}$ and instead learn parameters to perform adaptive scooping with our previously-defined bimanual scooping primitive, which performs each of the three phases in succession.

Given an initial overhead image observation $I_0$, CARBS starts by identifying the food position in the initial environment state $s_0$. To do this, we learn a segmentation model $f : \mathbb{R}^{W \times H \times C} \rightarrow \mathbb{R}^{W \times H}$ to obtain a food mask and food center position $(x_f, y_f)$ from the initial image $I_0$, which is then passed to our bimanual scooping primitive. As illustrated in Fig. 1, to differentiate between food deformabilities, we learn a Risk Classifier $r : \mathbb{R}^{W \times H \times C} \rightarrow \{0, 1\}$ that identifies an initial food image $I_0$ as "Robust" or "Fragile". In practice, we instantiate the classifier with a ResNet34 model [14] trained on a hand-labelled food fragility dataset with 600 images of 14 food classes. For robust foods, we set $\alpha = 1$ for maximum pushing of the food, since the food is not in danger of breaking and can benefit from the added pushing stabilization. For fragile foods, it is nontrivial to select $\alpha$ given only an initial observation $I_0$, so we propose a closed loop system for determining $\alpha$.

**Servoing for Fragile Foods.** At the beginning of each scooping rollout, we initialize $\alpha = 1$, indicating that the pusher should travel the full distance during the Pushing phase. However, varied food dynamics and differences in deformability can dictate the need for different $\alpha$ values even within the same food class. For example, when scooping two pieces of tofu, the food items may deform or slide on the plate differently due to food interactions with the plate, slight robot imprecision, or food shelf life. As a result, CARBS uses closed loop visual feedback in the form of a Failure Classifier $f : \mathbb{R}^{W \times H \times C} \rightarrow \{0, 1\}$ that identifies breakage-imminent states where the food item is in contact with the pusher and scooper, but not yet squeezed until breakage. This classifier is run at each state during Pushing (as in Fig. 1). When a breakage-imminent state is detected, the Pushing phase is terminated and $\alpha < 1$. For example, if a breakage-imminent state was detected after 65% of the Pushing phase had completed, the phase would terminate early and this would correspond

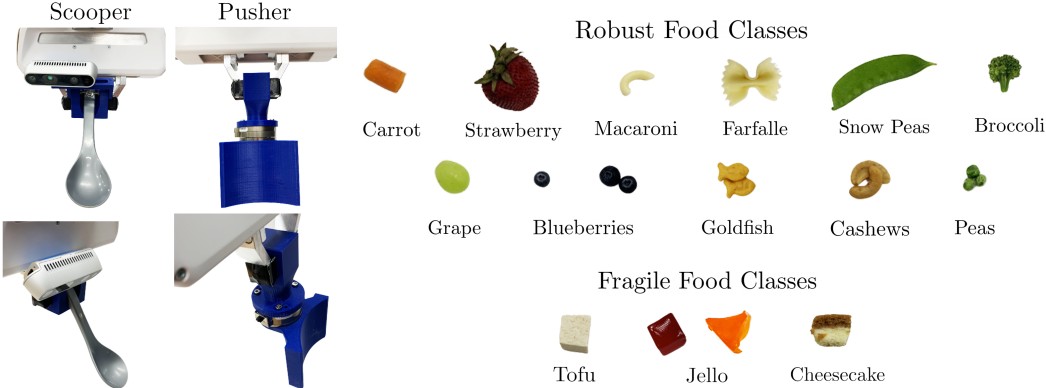

Figure 4: **Experimental Setup**: **Left**: The Pusher is a custom concave barrier used to push food items to the mouth of the spoon. The Scooper has an RGB camera mounted above the spoon to access images of the spoon mouth and surrounding workspace during scooping. **Right**: We consider scooping 16 food settings: 11 robust containing up to three food items, and 4 fragile.

to $\alpha = 0.65$ because the pusher had traveled 65% of the full pushing distance. We instantiate the Failure Classifier with a ResNet34 [14] model trained on hand-labelled images of 30 scooping trials.

## 4 Experiments

We validate CARBS's effectiveness on scooping foods of varied geometries and deformabilities. We design a series of experiments scooping 14 food items to demonstrate the advantage of a reactive bimanual strategy over hard-coded or single-arm actions. We select 14 food items to cover a wide range of sizes, shapes, and deformabilities: blueberry, broccoli, carrot, cashew, cheesecake cube, farfalle pasta, goldfish, grape, jello piece, macaroni pasta, pea, snow pea, strawberry and tofu cube (See Fig. 4). See Appendix A for food property details. We assume all food groups are pregrouped, where each food item is in contact with other items, if any, in the scene. Each set of food properties comes with a unique set of challenges. For example, blueberries, peas, and grapes are round, which may roll around the plate, while snowpeas and macaroni are irregularly shaped and can be difficult to stabilize within the spoon. We consider three deformable foods with diverse visual and material properties (jello, tofu, and cheesecake cubes), which are critically susceptible to breakage failures.

**Food Dataset.** The Segmentation model is trained on a dataset of 600 overhead RGB images of all food items except Orange Jello in the real workspace. We subtract a background image of the workspace to obtain masks of the food items. The Risk Classifier is trained on a the same dataset of overhead RGB images of food items. We hand-label each image as "Robust" or "Fragile", and augment 8X. The Failure Classifier is trained on images from scooping rollouts of *tofu only*, which we found to be sufficient for generalization to other food classes as well. More training details are in Appendix B. We collect a dataset by recording 60 image frames each of the Pushing phase of 30 rollouts with $\alpha = 1$, meaning we push the food the maximum distance. We then hand-label when the food item breaks in each rollout, and automatically generate labels per image as "Keep Going" or "Stop" for safe and breakage-imminent states respectively.

| Food Type | Success Rate | | |
|---|---|---|---|
| | *Single* | $\alpha = 1$ | *CARBS* |
| Broccoli | **5/5** | **5/5** | **5/5** |
| Grapes | 3/5 | **5/5** | **5/5** |
| Blueberry | 4/5 | **5/5** | **5/5** |
| Strawberry | **5/5** | **5/5** | **5/5** |
| Carrot | 4/5 | **5/5** | **5/5** |
| Farfalle | **5/5** | **5/5** | **5/5** |
| Macaroni | 2/5 | **5/5** | **5/5** |
| Snow Pea | 3/5 | **4/5** | **4/5** |
| Cashews (2) | 2/10 | **7/10** | **7/10** |
| Goldfish (2) | 6/10 | **8/10** | **8/10** |
| Blueberries (2) | **6/10** | **6/10** | **6/10** |
| Peas (3) | 3/15 | **15/15** | 14/15 |

Table 1: **Robust Food Physical Results:** We report the per food item success rate over 5 trials of scooping robust foods with the Single, $\alpha = 1$, and CARBS strategies. As expected, we observe CARBS matches $\alpha = 1$ performance across all robust foods. Both the CARBS and $\alpha = 1$ methods match or outperform the Single baseline, suggesting that the bimanual stabilizing strategies (Angled Pusher, Cupping Motion, and Pinning Motion) are advantageous over a static barrier.

**Implementation Details.** Our real-world environment setup consists of two 7-DoF Franka Emika Panda arms, each holding a Scooper or Pusher tool as shown in Fig. 4. The robot bases are set to be parallel with each other with the plate workspace between two bases, and both robots are controlled with an impedance controller. The Scooper is mounted at a 45 degree angle with an angled Intel Realsense D435 camera above the spoon as shown in Fig. 4. We designed a custom 3D printed pusher with a concave surface to encourage food grouping and stabilization. See Appendix C for a discussion of design choices for the pusher and the spoon. Our failure classifier to servo for breakage runs at 20 Hz frequency.

| Food Type | Avg. Weight Difference (%) | | | | $\alpha$ Value |
|---|---|---|---|---|---|
| | *Single* | $\alpha = 0.93$ | $\alpha = 1$ | *CARBS* | |
| Tofu | 23.194 | 41.099 | 2.474 | **0.444** | 0.9477±0.019 |
| Red Square Jello (Failure OOD) | 40.700 | 21.376 | 1.076 | **0.420** | 0.9400±0.021 |
| Cheesecake (Failure OOD) | 26.873 | 25.937 | 10.867 | **6.639** | 0.9231±0.016 |
| Orange Triangle Jello (CARBS OOD) | 100 | 62.274 | 9.720 | **0.449** | 0.9169±0.030 |

Table 2: **Fragile Food Physical Results:** We report the weight loss of food items after scooping as a percentage of the original food weight, averaged across 5 trials. We also report the average values of CARBS's stabilizing parameter $\alpha$, a scaling value for the total pushing distance (13 cm) with a 95% confidence interval. We observe that CARBS's Failure Classifier adjusts $\alpha$ to end pushing at a different position depending on the food. This suggests that the classifier learns to recognize the bimanual constraint that leads to breakage rather than a fixed ending position for all fragile foods. We also find that the triangle jello has the highest $\alpha$ variability, possibly due to its irregular shape relative to the cube foods. We observe CARBS's Failure Classifier generalizes to novel fragile food classes (Jello, Cheesecake) and its Risk Classifier generalizes to varied visual appearances and geometries within one class (Orange Jello).

**Baselines.** We compare against three baselines: **Single**, $\alpha = 0.93$ and $\alpha = 1$. **Single** executes a single-arm scooping method where the pusher is fixed and acts as a static barrier. The spoon moves towards the food and pushes the food against a stationary pusher during scooping. Notably, the pusher does not use any of the bimanual stabilizing strategies shown in Fig. 3. $\alpha = 0.93$ executes a bimanual scooping primitive without an adaptive stabilizing strategy where the pushing distance is **0.93** of the total 13cm pushing distance, stopping approximately 1cm early to prevent breakage. $\alpha = 1$ executes a bimanual scooping primitive instead with the full 13cm pushing distance ($\alpha = 1$), which is identical to the primitive for scooping a robust food item.

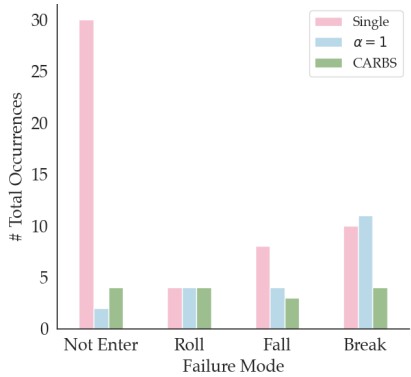

Figure 5: **Failure Modes**: We observe 4 failure modes across scooping strategies: (Not Enter) foods contact the scooper but do *not enter* the spoon bowl, (Roll) foods *roll out* of scooping range, (Fall) foods *fall out* of the spoon after being scooped into the spoon, and (Break) *breakage*. There are many (Not Enter) failures with the Single baseline because irregularly shaped foods and multiple items are difficult to roll into the spoon with a static pusher. We also find both baselines have a higher occurrence of breakage failures, supporting the need for an adaptive stabilizing strategy.

**Results.** We compare our method, CARBS, against two baselines across 14 different food items, with 11 robust foods and 4 fragile foods. We also include an additional $\alpha = 0.93$ baseline for the 4 fragile foods, which is the average $\alpha$ value across all CARBS fragile food trials. For robust foods, we report binary scooping success as whether the food ended within the spoon bowl after scooping in Table 1. We consider settings of single foods of varied geometries, and additional settings of up to 3 food items. As expected, CARBS and $\alpha = 1$ have similarly high performances because CARBS should learn to set $\alpha = 1$ for Robust foods. We compare to the Single baseline to observe the advantages of the three bimanual scooping strategies (Angled Pushing, Cupping, and Pinning as in Fig. 3) over a static pusher position. For round foods, the Angled Pusher stabilization (described in Sec. 3) is important for building momentum and helping the item roll into the scooper, which prevents the food from rolling away (Roll Failure). For irregularly shaped foods that often extend past the spoon edges, the food items can be in unstable poses even once in the spoon bowl, which can cause them to fall out during the Scooping phase. These items benefit

from the Pinning stabilizing strategy to pin the food in place throughout the rotation motion and prevent them from falling out (Fall Failure). See Appendix F for an ablation study on the stabilizing strategies.

Lastly, we consider scooping 2-3 food items simultaneously, as inspired by the example of chasing peas around a plate. While CARBS and the $\alpha = 1$ baseline still outperform the Single baseline, they fail to achieve as high success as scooping a single food item. It is more difficult to stabilize multiple food items simultaneously due to the added dynamics complexity. For example, stabilizing two blueberries to ensure *both* roll into the mouth of the scooper is nontrivial (Not Enter Failure). The food dynamics also become more complicated as multiple foods can interact not only with the pusher and scooper tools, but also with each other.

We present experiments scooping four fragile food settings in Table 2. Two food settings are out of distribution for our Failure Classifier and one is out of distribution for both the Failure and Risk Classifier. We report the weight loss during scooping as a percentage of the original food weight to measure the breakage failure severity. CARBS is able to reduce food breakage by 16.185% compared to the $\alpha = 1$ baseline. This suggests that the Failure Classifier can effectively recognize breakage-imminent states and adapt the stabilizing parameter $\alpha$ to prevent breakage. We also find that $\alpha = 0.93$ baseline, with a fixed early stop of 1cm to prevent breakage, has worse performance than CARBS because the fixed distance cannot adapt to different food geometries and properties. Some smaller foods fail to fully enter the spoon bowl due to the early stop and are not scooped, resulting in 100% weight loss. We note that although the average $\alpha$ value for cheesecake is 0.9231, there is still a large gap in performance between CARBS and the $\boldsymbol{\alpha = 0.93}$ baseline because CARBS is able to adjust $\alpha$ for each food item. We additionally report the 95% confidence intervals for the CARBS $\boldsymbol{\alpha}$ values to highlight this adaptability. We note that the weight difference for CARBS for cheesecake, while still lower than the Single and $\alpha = 1$ baselines, is significantly higher than tofu and jello. This is due to the stickiness of the cheesecake and its propensity to leave food residue on the plate and tools during movement. We also report the $\alpha$ values learned with CARBS and show that although our Failure Classifier is only trained on one food class (tofu), CARBS is able to adjust $\alpha$ across novel food classes depending on their shape, size, and deformability. This supports our claim that learning to detect failures from vision generalizes across breakage-prone food classes. We also find CARBS's Risk Classifier generalizes within a food class to varied visual appearance and geometries, suggesting the effectiveness of CARBS for scooping novel foods.

## 5   Discussion

**Summary.** We present CARBS, a learned bimanual scooping policy for robustly scooping food items of varied geometries and deformabilities. CARBS learns a dynamic stablizing strategy to avoid breakage failures when scooping high-risk foods by identifying breakage-immenent states and adjusting the stabilizing action parameter: the distance between the scooper and pusher. We evaluate the generalizability of CARBS with physical experiments scooping 14 foods of varying shapes, sizes, and fragility, and compare against two baselines. We find that CARBS is able to successfully scoop 85.7% of foods.

**Limitations and Future Work.** CARBS struggles to scoop foods with uncommon material properties and complex dynamics, and multiple food items. While our system is able to reduce cheesecake breakage compared to baselines, it does not achieve similar success compared to other fragile foods due to the cheesecake's stickiness. CARBS also struggles when scooping multiple blueberries with unpredictable dynamics. Their round shape and inertia allow them to roll off not only the scooper and pusher, but also each other. CARBS leaves room for improvement when scooping multiple food items as well – it is nontrivial to determine an optimal stabilizing policy for multiple items at once. In future work, we hope to study more dynamic stabilizing strategies for food acquisition and other bimanual tasks, such as tying knots and buttoning clothes. We plan to relax our food environment assumptions and scoop an even larger range of foods, for example by pushing to group multiple scattered peas on a cluttered plate and then scooping into a spoon. These cluttered food settings require longer horizon planning using potentially new primitives to group then acquire the food, which we leave to future work. We will also explore multimodal sensing strategies with new probing primitives for scooping to generalize to more unseen foods and augment our vision-only system.

## Acknowledgments

This project was sponsored by NSF Awards 2132847, 2006388, and 1941722, and the Office of Naval Research (ONR). Jennifer Grannen is further grateful to be supported by an NSF GRFP. Any opinions, findings, conclusions or recommendations expressed in this material are those of the authors and do not necessarily reflect the views of the sponsors. We additionally thank our colleagues who provided helpful feedback and suggestions, especially Priya Sundaresan.

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
