# OpenReview forum: "Learning Bimanual Scooping Policies for Food Acquisition"
_robot-learning.org/CoRL/2022/Conference — CoRL 2022 Poster_

### Official Review · Reviewer_rkd9 · 2022-07-29

**Originality:** Good
**Technical Quality:** Good
**Clarity Of Presentation:** Good
**Impact:** 2

**Recommendation:**

Weak Accept: I recommend accepting the paper, but will not argue for my recommendation if the majority of other reviewers have a different opinion.

**Summary:**

The authors proposed a general bimanual scooping primitive and an adaptive stabilization strategy that enables successful acquisition of a diverse set of food geometries and physical properties. The author's approch, CARBS learns to stabilize without impeding task progress by identifying high-risk foods and robustly scooping them using closed-loop visual feedback. The robot takes two inputs, pusher travel distance and food posiiton, and performs three stabilizing strategies. The Risk Classifier classifies food fragility into binary categories (robust or fragile), and the Failure Classifier determines whether the robot is "Keep Going" or "Stop" and adjusts the stabilizing parameter to prevent the object from being broken. With these two classifiers and the three strategies described above, CARBS generalize across food shape, size, and deformability and is additionally able to manipulate multiple food items simultaneously.

**Issues:**

I criticized this study as engineering, but it was wrong, and I understood that this work accomplished after a great deal of preliminary study. The authors' experimental results contain a many important findings for the entry of robots into the home and food industry.

**Quality Of The Limitations Section:**

Additional details required

**Reviewer Expertise:**

4: The reviewer is confident but not absolutely certain that the evaluation is correct

**Robotics Focus:**

Sufficient demonstration on hardware

**Strengths And Weaknesses:**

The authors address a very challenging task of food scooping, a subject for which there is little previous research. The supplemental video fully conveyed the appeal of the study and further helped readers understand it. The fact that the twin-armed robot performs a difficult task is also highly commendable; one can read from the paper that considerable trial and error went into finding the three stabilization strategies. From the comparison experiment, the reviewers understood the effectiveness of the proposed method as the weight loss of food was significantly lower than the baseline.

However, the technical novelty of the proposed method is low; it is an engineering combination of existing technologies. The three stabilization strategies (1) Angled Pusher states that the Pusher was placed at a specific angle, but this is an easy thing to come up with and not a point of special claim. If the angle changes dynamically according to the shape and characteristics of the object, it can be broadly claimed as a novelty. The Planning of the Scoping Phase is also the same.

There is also a lack of specific explanation of the methodology, which confuses the reader. Since the whole system configuration is difficult to understand, it is better to show the following three in Fig. 1 to help the reader understand: (1) the name of the neural network model used, (2) the name of the information output by each module (e.g., the food position) are shown above the arrows, and (3) the data flow (steps) at runtime. In line 247, the data augmentation is used to trained the classifier, but what kind of method was used specifically?
Since there is no description of the operation cycle during task execution, it is unclear how often the Failure Classifier performs object status determination. In line 247, "The Failure Classifier is trained on images from scooping rollouts of tofu only, which we found to be sufficient for generalization to other food classes as well," requires quantitative data. The reviewer doubts that generalization can be achieved by simply learning tofu image data, since the foods have very different shapes, colors, characteristics (robust or fragile), and dynamics (e.g., blueberries with unpredictable dynamics).

Minor comments are as follows.
Line 178: Which direction does the x-axis indicate in Fig. 3? Showing each axis in the figure will help the reader understand. The text also needs to be improved, as there are many typographical errors. e.g. Appendix reference number and Figure3/Fig.3. Since the reference to Fig. 3 appears four times between lines 179 and 188, it is easier to understand by adding sub-numbers like Fig. 3 (a) and (b). How is the generalization performance of Risk Classifier? The reviewer is concerned that it may be limited because there are many similar looking but different hardnesses in the world. As the author describes in the limitation section, it is better to build inference models based on force sensors and tactile sensation. Multimodal learning allows for highly accurate estimation of contact states.

**Summary Of Recommendation:**

The task is challenging and the video is well presented.

---

> ### Author Response · Authors · 2022-08-22
> **Response to Reviewer rkd9**
>
> **Comment:**
>
> We thank the reviewer for their time and thoughtful feedback. We appreciate the reviewer for their acknowledgement of the difficulty of bimanually scooping foods and the effort required to build this system. We emphasize that this work is intended to be a systems paper, and we believe any robust system requires combining pre-existing engineering, design, and learned components together to solve a novel problem. We strongly believe these components are necessary and non-trivial in developing robust and general systems that can make an impact and require more attention in our research papers. We have incorporated their feedback and clarifications into the paper in blue text to make the writing more clear and comprehensive, and hope that our responses address the reviewer’s concerns.
>
> > However, the technical novelty of the proposed method is low; it is an engineering combination of existing technologies.
>
> As mentioned earlier, we do not claim that any of our individual system components are technically novel, but rather propose a full system that requires integratinging learning algorithms along with non-trivial engineering and design choices to form a *novel system* that solves the challenging problem of scooping a variety of foods. To the best of our knowledge, our work is the first to consider the both tasks of bimanual food acquisition and scooping a wide assortment of food items, varying in visual appearance, size, geometry, and deformability. We strongly believe any robotics systems capable of robustly performing novel tasks — such as CARBS capable of scooping real foods — require integration of design and engineering choices beyond just algorithmic contributions, and in our opinion that is an important and necessary contribution in advancing the field of robotics.
>
> > The three stabilization strategies (1) Angled Pusher states that the Pusher was placed at a specific angle, but this is an easy thing to come up with and not a point of special claim.
>
> In our preliminary experiments, it was not initially obvious to us what the stabilizing strategies should be. In fact, we converged to these strategies after experimenting with 25 numbers of different food items with varying appearance, size, geometry, and deformability. And only after widening our space of scooping tasks, we concluded on the specific stabilizing strategies described in this paper. We understand that many of these after the fact might seem not special, but the process of finding them was definitely not trivial (at least for us!)
>
> To further demonstrate the importance and necessity of each stabilizing strategy, and address the reviewer’s concerns, we present new additional ablations of the stabilizing strategies in Appendix F, along with videos of motivating failure cases for each attached to this response. We find that without the Angled Pusher strategy, irregularly-shaped, robust foods like cashews may become wedged between the spoon and pusher and spring out of the scene. Without the Cupping strategy, foods do not build momentum to enter the spoon, and as a result may be broken as shown with this deformed blueberry video. Lastly, without the Pinning strategy, irregularly-shaped foods often fall out of the spoon, as shown with the macaroni failure.
>
> From these experiments, we hope the reviewer agrees that the three stabilizing strategies are necessary for scooping such a broad range of foods. We believe defining these can also be potentially useful for other bimanual tasks that require close coordination between two arms in a similar way as bimanual scooping.
>
> > If the angle changes dynamically according to the shape and characteristics of the object, it can be broadly claimed as a novelty. The Planning of the Scoping Phase is also the same.
>
> We thank the reviewer for this interesting suggestion. While changing the spoon angle was something we initially considered when defining the parameters of our scooping primitive, we found that a single spoon angle was able to successfully acquire all the robust food items. Because we did not want to unnecessarily complicate our scooping primitive, we did not include this spoon angle parameter in our definition. We consider the simplicity of our primitive to be a strength rather than a weakness, as this means it is easier to implement and is more likely to be adopted by future work. We hope the reviewer agrees with us.
>
> > ... it is better to show the following three in Fig. 1 to help the reader understand: (1) the name of the neural network model used, (2) the name of the information output by each module (e.g., the food position) are shown above the arrows, and (3) the data flow (steps) at runtime.
>
> Thank you for your helpful suggestions. We apologize if the initial figure was not clear. We have included in our updated version an updated Fig. 1 with labeled data flows for each module and at runtime, along with the network names. We have also separated the two scooping rollouts in Fig. 1 for clarity.
>
>
> **Zip File:**
>
> /attachment/313d7b3a54e2bf05d837eb885474bb3162ac2270.zip

---

> ### Author Response · Authors · 2022-08-22
> **Response to Reviewer rkd9 (Part 2)**
>
> (continuing due to space limits)
>
> > the data augmentation is used to trained the classifier, but what kind of method was used specifically?
>
> We use the following image transformations from the imgaug library[1] to perform all data augmentations: Linear Contrast, Add, Gamma Contrast, Gaussian Blur, Multiply Saturation, and Additive Gaussian Noise. We have added examples from the augmented dataset in Appendix B, along with a table describing the parameters for each augmentation.
>
> > it is unclear how often the Failure Classifier performs object status determination.
>
> We servo at a frequency of 20 Hz. We have added this to the paper at L260.
>
> > In line 247...The reviewer doubts that generalization can be achieved by simply learning tofu image data, since the foods have very different shapes, colors, characteristics (robust or fragile), and dynamics (e.g., blueberries with unpredictable dynamics)
>
> We understand the reviewer’s skepticism. Let us further explain the approach: The failure classifier is only deployed on foods that are fragile because the risk classifier identifies the robust foods, which do not require servoing for breakage. As a result, the failure classifier does not need to generalize to robust foods. Additionally, fragile foods tend to have similar breakage failures and these breakages are due to their similar dynamics and properties, which allows for ease of generalization. We hypothesize that the failure classifier is able to generalize because it learns to recognize when the food object comes into contact with both the pusher and the spoon, rather than overfitting to tofu.
>
> We evaluate this failure classifier on both tofu (which it was trained on) and three additional out of distribution foods of different shapes, colors, and deformabilities (orange jello triangle, red jello cube, and cheesecake cube). In Table 2, we find that this failure classifier is able to successfully generalize as CARBS is able to reduce food breakage for all out of distribution food classes. Additionally, we find the failure classifier is able to accurately classify breakage in 96.8% of 157 images over 13 rollouts of the three out of distribution foods, which we have added to the paper as well at Line 474-476.
>
> > Which direction does the x-axis indicate in Fig. 3? Showing each axis in the figure will help the reader understand
>
> We thank the reviewer for this suggestion, we have added the axes to Fig. 3.
>
> > The text also needs to be improved, as there are many typographical errors. e.g. Appendix reference number and Figure3/Fig.3
>
> Thank you for drawing our attention to this. We have fixed the references and typographical errors in our updated version.
>
> > Since the reference to Fig. 3 appears four times between lines 179 and 188, it is easier to understand by adding sub-numbers like Fig. 3 (a) and (b)
>
> This is a helpful suggestion, thank you! We have added sub-numbers into Fig. 3.
>
> > How is the generalization performance of Risk Classifier? The reviewer is concerned that it may be limited because there are many similar looking but different hardnesses in the world. As the author describes in the limitation section, it is better to build inference models based on force sensors and tactile sensation. Multimodal learning allows for highly accurate estimation of contact states
>
> We show generalization across food color and shape with the orange jello triangle experiment in Table 2, which is not present in the Risk Classifier training data. However, we agree that generalizing to entirely new food classes may be difficult without prior knowledge about the food. For the generalization problem setting, including tactile information would be helpful for probing the food to determine fragility before scooping, especially those foods that are visually similar but have different deformabilities.
>
> In Appendix H, we report force/torque readings from a sensor mounted on the pusher tool. While prior work [2] has used tactile sensing for food skewering, we find that the readings are too noisy to distinguish between robust and fragile food classes. This is because skewering a food object is an isolated interaction that yields clean force readings, whereas the scooper and pusher tools scraping against the plate surface add too much noise to the readings. We highlight this in the Limitations section on Line 358-360, and leave the incorporation of tactile sensing for scooping to future work.
>
> We again thank the reviewer for their helpful comments that we have incorporated to improve the paper clarity. We are happy to address any additional concerns or questions. In the light of our responses, additional experiments, and clarifications, we ask the reviewer to consider raising their score.
>
> [1] https://imgaug.readthedocs.io/en/latest/
> [2] E. Gordon, S. Roychowdhury, T. Bhattacharjee, K. Jamieson, and S. S. Srinivasa. “Leveraging Post Hoc Context for Faster Learning in Bandit Settings with Applications in Robot-Assisted Feeding” ICRA 2021.

---

> ### Author Response · Authors · 2022-08-25
> **Follow Up to Response to Reviewer rkd9**
>
> We hope our new experiments and clarifications have adequately addressed your concerns about our work. Because there is only one day left in the rebuttal period, we are following up to see if you have chosen to raise your rating or if there is anything we can do to have you reconsider before the rebuttal period ends.

---

### Official Review · Reviewer_aovx · 2022-07-30

**Originality:** Very Good
**Technical Quality:** Very Good
**Clarity Of Presentation:** Excellent
**Impact:** 4

**Recommendation:**

Strong Accept: I recommend accepting the paper and will argue for my recommendation even if other reviewers hold a different opinion.

**Summary:**

This paper proposes a method for scooping foods into a spoon in which both the spoon and a "pusher" are attached to separate robot arms. The authors show that allowing independent motion for the spoon and pusher significantly improves success rates vs a spoon-only strategy. The authors also use two neural networks as part of their control architecture: a risk classifier that decides whether to use a fixed pushing strategy for "robust" foods or an adaptive strategy for fragile foods, and a failure classifier that allows the robot to predict imminent breakage of fragile foods and adapt the relative usage of the pusher and spoon accordingly. The authors provide data from hardware experiments showing that their approach improves both over single-arm strategies and non-adaptive strategies (for fragile foods).

**Issues:**

- As mentioned above, more detail about how the parameter $\alpha$ is adapted would be helpful.
- You define the image observations as the state of an MDP, but I usually think of images as observations rather than states, suggesting that a POMDP could be a better fit here. This aligns with the fact that you use a learned state estimator to get the centroid of the food from the image, suggesting that there is an underlying state that can be inferred from the image observations. Perhaps making this adjustment would help you expand to multimodal sensor feedback (as you mention in the limitations section).

**Quality Of The Limitations Section:**

Limitations are addressed clearly

**Reviewer Expertise:**

3: The reviewer is fairly confident that the evaluation is correct

**Robotics Focus:**

Sufficient demonstration on hardware

**Strengths And Weaknesses:**

## Strengths

- This paper includes very impressive hardware results; nice job!
- I think that the architecture in this paper strikes a good balance between hand-designed and learned components, as opposed to an entirely learned policy.

## Weaknesses

- The paper is a bit light on the details of the scooping primitives. Particularly, it took a couple of takes to understand the role of $\alpha$, and I'm still not quite sure how $\alpha$ is adaptively varied. Perhaps adding a section in your supplementary material would help with this point.

**Summary Of Recommendation:**

This paper was clear and enjoyable to read, and the authors present fairly comprehensive hardware results to demonstrate the effectiveness of their method.

---

> ### Author Response · Authors · 2022-08-22
> **Response to Reviewer aovx**
>
> We thank the reviewer for their time and helpful comments. We appreciate the reviewer’s comment on CARBS being “a good balance between hand-designed and learned components”, as scooping foods is a complex manipulation task that requires both engineering and learned elements.
>
> > The paper is a bit light on the details of the scooping primitives. Particularly, it took a couple of takes to understand the role of $\alpha$, and I'm still not quite sure how $\alpha$ is adaptively varied. Perhaps adding a section in your supplementary material would help with this point.
>
> We thank the reviewer for pointing out this point of confusion in the paper. $\alpha$ is a parameter that indicates the distance traveled by the pusher during the Pushing phase. When $\alpha = 1$, the pusher travels the full distance from its initial position until it touches the spoon mouth at $(x_f + d, y_f)$. When fragile foods are at risk for breakage, the Pushing phase will terminate early when the Failure Classifier detects a breakage-imminent state. As a result, the pusher does not travel the full distance, and $\alpha$ is < 1. We have added a more detailed explanation of $\alpha$ in Section 3 of the paper in blue on Lines 217-229.
>
> > You define the image observations as the state of an MDP, but I usually think of images as observations rather than states, suggesting that a POMDP could be a better fit here… limitations section)
>
> Thank you for pointing this out, we completely agree and have updated the text to change our model from MDP to POMDP to capture the fact that images are partial observations of the true state of the environment. This is written in colored text in the revised version on Lines 156-157.
>
> We hope the above explanation answered the reviewer’s question about $\alpha$, and once again very much appreciate their feedback.

---

> > ### Comment · Reviewer_aovx · 2022-08-22
> > **Reponse to comment**
> >
> > Great; thank you.

---

### Official Review · Reviewer_1gVW · 2022-07-31

**Originality:** Very Good
**Technical Quality:** Very Good
**Clarity Of Presentation:** Very Good
**Impact:** 4

**Recommendation:**

Weak Accept: I recommend accepting the paper, but will not argue for my recommendation if the majority of other reviewers have a different opinion.

**Summary:**

This work focuses on bimanual scooping of single instances of that would be difficult to skewer or scoop using traditional techniques due to their unique geometries, i.e. objects that are small, brittle, fragile, or those that would roll away if scooped with a single arm. To handle these problematic foods, a second arm with a pushing attachment is used to stabilize, collect and push the food onto the scooping attachment on the other arm. Three primitives are defined; angled pushing, adaptive cupping, and pinning. Adaptive cupping has a variable parameter, \alpha, that defines how close to the scooper the pusher gets, as well as the location of the center of the food item. These parameters are learned using a convolutional network, based on images of the food. The centers and segmentation of each food are predicted as well as whether the food is robust or fragile. If it is fragile, a breakage detection prediction is run online and the food is pushed until imminent breakage is detected.

**Issues:**

The parameter in of the angled pushing primitive is described but never defined. How is \theta selected? Is it constant for all foods? The spoon angle is described in the supplemental material but not the pusher angle.

Most food acquisition tasks are done on relatively cluttered plates. Some experiments showing robustness to this or mention of it in the limitations section should be added.

Figure captions are too close to text, particularly above line 207.


**Quality Of The Limitations Section:**

Additional details required

**Reviewer Expertise:**

4: The reviewer is confident but not absolutely certain that the evaluation is correct

**Robotics Focus:**

Sufficient demonstration on hardware

**Strengths And Weaknesses:**

**Strengths**

_Novel Solution to an Important Problem:_

This is a novel solution to a previously unsolved problem. Further, this is a problem that is applicable to people who use robotics feeding systems. The solution was fully implemented into a working system that was tested on real food.

_Simple and Effective Solution:_

The solution uses simple primitives that are adapted to individual food instances. The pusher design and primitives leverage funneling behaviors that reduce uncertainty in the position of the food, improving the probability of success. Further, relatively simple representation of the food allows the systems to generalize across types of food. The method shows significant improvement over the static baseline, showing that some dexterity on the second arm is necessary. Further, the closed loop stopping behavior greatly reduces food breakage.

**Weaknesses**

_Learning components are relatively limited:_

The learned components of this method are relatively limited. They amount to a food segmenter and centroid predictor, food robustness classifier, and a food breakage predictor. While these are sufficient to complete this task with a high degree of efficiency, they do not showcase the need for learning in robotics tasks.

_Could use more complex baselines:_

The baselines were somewhat limited, with a static pusher, a constant \alpha = 1, and the proposed method. A method that used the predicted centroid of the food, stopping when the centroid was over the spoon, could show the need for a breakage specific online policy. Additionally, a lower \alpha baseline, or set of constant \alpha baselines could show that no single \alpha value works across food types.

**Minor Weaknesses**

The parameter in of the angled pushing primitive is described but never defined. How is \theta selected? Is it constant for all foods? The spoon angle is described in the supplemental material but not the pusher angle.

Most food acquisition tasks are done on relatively cluttered plates. Some experiments showing robustness to this or mention of it in the limitations section should be added.

**Summary Of Recommendation:**

This work shows a simple application of learning to solve an important, previously unsolved problem. While the learned components are relatively simple, I believe that they are sufficient to solve the problem. Additionally, the work shows a full system, working with real word food, in a relatively realistic manner (minus the lack of clutter). If the conference would like to showcase more realistic systems, using some learning to solve relevant tasks, I believe this paper is an example of such work.

---

> ### Author Response · Authors · 2022-08-22
> **Response to Reviewer 1gVW**
>
> We thank the reviewer for their time and insightful suggestions. We thank the reviewer for their description of our system, CARBS, as a “simple and effective solution” to the “important problem” of food manipulation and acquisition. We provide the following responses to discuss the weaknesses and concerns of the reviewer, and have incorporated all changes into the attached paper version in blue.
>
> > Learning components are relatively limited.
>
> In this work, we have definitely benefited from learning, but we have used learning in a measured way. We found a combination of system design along with learning together was the most effective way of designing a system that accomplishes bimanual scooping.  We were hesitant to incorporate additional learning components when it was not necessary. We view the simplicity of our system as a strength rather than a weakness, as it will be easier for future works to implement and adopt CARBS without compromising on scooping performance.
>
> > Could use more complex baselines… A method that used the predicted centroid of the food, stopping when the centroid was over the spoon, could show the need for a breakage specific online policy.
>
> The centroid position baseline is an interesting heuristic for breakage. However, the food item often moves during the angled pushing phase and therefore the initial centroid position is often not accurate throughout the scooping motion. We could instead choose to servo for an updated centroid position during scooping. However, we decided to not include this baseline because this is a strictly harder task than simply servoing with a failure classifier model and we do not need the additional food position information.
>
> > Could use more complex baselines… Additionally, a lower \alpha baseline, or set of constant \alpha baselines could show that no single \alpha value works across food types.
>
> Thank you for this suggestion. We agree, showing that no single $\alpha$ value works across food types is a valuable result to show. We have added an additional baseline of a constant $\alpha = 0.93$ to Table 2 to show that a single $\alpha$ value < 1 is still insufficient for successfully scooping while preventing breakage of all fragile food classes. In Table 2, we additionally report the $\alpha$ values for each fragile food class and find that these values indeed vary by up to 0.03 which translates to approximately a 0.4cm range.
>
> > The parameter in of the angled pushing primitive is described but never defined. How is \theta selected? Is it constant for all foods?
>
> $\theta$ was empirically selected to be a constant 15 degrees for all foods. This value accomplished our desired behavior of foods rolling into the spoon while avoiding the problem of foods becoming wedged and stuck under the pusher surface. We have added this information to the main text where $\theta$ is introduced on Line 185.
>
> > Most food acquisition tasks are done on relatively cluttered plates. Some experiments showing robustness to this or mention of it in the limitations section should be added.
>
> Thank you for pointing this out. We agree, most food acquisition settings are cluttered and this is an impactful application that we hope to tackle next. Scooping in clutter requires using different and potentially hierarchical action primitives such as grouping foods on a plate before attempting the acquisition primitives, and presents a different set of planning challenges to organize and acquire foods on a plate. This is a longer horizon task, and we agree this is an interesting problem. Our work does not address this problem setting, but instead focuses on the scooping action alone and studies the components needed to successfully scoop a wide variety of foods. A full scooping system on a cluttered plate could employ CARBS, our scooping system, for acquiring foods once they are organized. While we do mention in the limitations section that we hope to relax our food environment constraints in the future, we have further emphasized that specifically cluttered settings are important to consider next (L354-358).
>
> We once again thank the reviewer for their comments and have updated the attached paper version with these notes.

---

> > ### Comment · Reviewer_1gVW · 2022-08-26
> > **Baselines and constant alpha**
> >
> > >The centroid position baseline is an interesting heuristic for breakage. However, the food item often moves during the angled pushing phase and therefore the initial centroid position is often not accurate throughout the scooping motion. We could instead choose to servo for an updated centroid position during scooping. However, we decided to not include this baseline because this is a strictly harder task than simply servoing with a failure classifier model and we do not need the additional food position information.
> >
> > Even if this fails, I think its good to show that its not trivial to solve the task in this way.
> >
> > >Thank you for this suggestion. We agree, showing that no single  value works across food types is a valuable result to show. We have added an additional baseline of a constant  to Table 2 to show that a single  value < 1 is still insufficient for successfully scooping while preventing breakage of all fragile food classes. In Table 2, we additionally report the  values for each fragile food class and find that these values indeed vary by up to 0.03 which translates to approximately a 0.4cm range.
> >
> > It is very interesting that this fails so spectacularly. This looks like evidence that dynamically predicting this value is necessary to solve this task. I would add that the value .93 was selected. I assume its the average across all foods.

---

> > > ### Author Response · Authors · 2022-08-26
> > > **Centroid Position Baseline**
> > >
> > > Thank you for this suggestion -- we have started running scooping experiments with this centroid baseline method. Unfortunately, we will not be able to finish running these experiments before the end of the rebuttal period tomorrow, but will be sure to report these results in the final camera-ready version of the paper if accepted.
> > >
> > > We have also reported the specific baseline value of $\alpha = 0.93$ in our paper version, and have included the detail that it is the average $\alpha$ value across all fragile food trials.

---

> ### Author Response · Authors · 2022-08-25
> **Follow up with Response to 1gVW**
>
> We hope the new baseline experiments and theta information in our updated paper version have addressed your concerns. Because there is only one day left in the rebuttal period, we are following up to see if you have any additional feedback or suggestions we can discuss to improve our paper.

---

### Official Review · Reviewer_FpM9 · 2022-08-06

**Originality:** Good
**Technical Quality:** Good
**Clarity Of Presentation:** Good
**Impact:** 3

**Recommendation:**

Weak Accept: I recommend accepting the paper, but will not argue for my recommendation if the majority of other reviewers have a different opinion.

**Summary:**

This paper introduces a bimanual scooping primitive that is capable of stably scooping bite-size food items with different geometry, color and deformation properties. In this case the stability of the scoop refers to avoiding breakage, rolling or cutting of the food item being scooped. The primitive, formulated as an MDP, takes as input an RGB image of a camera mounted on the scooping arm and selects the stabilizing strategy to perform the scoop either: angled push, cupping or pinning motions. For fragile food items an adaptive cupping strategy is introduced that leverages a “risk classifier” trained to identify high-risk breakage prone foods and a “failure classifier” trained to identify near-breakage states to adapt the cupping motion by scaling the distance traveled between the pusher and the scoop. Improved performance over single-arm scooping and baseline bimanual scooping (no active pusher) is shown.

**Issues:**

Comments listed in the weaknesses section should be addressed.

**Quality Of The Limitations Section:**

Limitations are addressed clearly

**Reviewer Expertise:**

5: The reviewer is absolutely certain that the evaluation is correct and very familiar with the relevant literature

**Robotics Focus:**

Sufficient demonstration on hardware

**Strengths And Weaknesses:**

**Strengths**

- The paper is tackling an interesting and difficult problem, as food manipulation is not subject to a single form of deformation or failure, this paper seems to solve a range of issues that can arise from scooping food items with different properties.
- Convincing videos and experiments.
- Catchy acronym.

**Weaknesses**

- It is not entirely clear to me what is being learned in the MDP and how that data is acquired. In section 3 the MDP is introduced and states that the transition dynamics are unknown, if they are unknown then they must be learned or parametrized somehow? Also, first it is mentioned that the action space is 14-DoF but then it is mentioned that it is simplified by the bimanual primitive which has 3 phases, so is the new action space 3 dimensions? This is not really clarified in the main text or appendix.
- The paper only mentions the learning pipeline for the risk and failure classifiers, however, shouldn’t there be a dataset where the robot learns the transitions between the different scooping phases/stabilizing strategies? For example, from angled pushing to cupping to pinning. It is unclear to me if this sequence or transitions between the phases are learned from the food interactions or if they are predefined and parametrized by the designers. How is the MDP deciding when to transition between phases?
- Related to the previous point, the paper only gives details about the risk $r:\mathbb{R}^{..}\rightarrow$ \{0,1\} and failure classifiers  $f:\mathbb{R}^{..}\rightarrow${0,1} which both seem to map to a binary classification variable. It is not clear to me how the outputs of these functions are used to modulate $\alpha$ which seems to be a continuous function based on the figures and videos.
- It would be ideal if the authors included a baseline comparison to the work cited from [23] which learns the bimanual interaction constraints from human demonstrations. This would clarify the doubts related to what is being learned and what is assumed/predefined in this framework.
- Is there a reason why adaptation to avoid breakage is only considered in the cupping phase? It would seem to me that breakage could be possible during the angled pushing and pinning phases as well - to avoid breakage or rolling there you would need to adapt the angle used by the pusher or modulate the speed of the rotation done by the pinning motion.
- It would be interesting to see how the primitive performs when trying to scoop items that are slightly larger than the scoop itself, like 2 strawberries or a larger piece of jello or cheesecake, oftentimes in the kitchen we use tools that are not the appropriate size but as humans we adapt our motion to perform the task, could this form of adaptation be translated to robot through this adaptive scooping framework?
- While the proposed primitive seems appropriate for scooping it would be great if the authors discussed how the presented strategy can be transferred to other types of tasks or if it can be adapted to handle other types of items that are not food but need a scooping mechanism like soil, sand, etc.
- Details are lacking regarding the low-level control of the arms, is the control done purely in position mode with visual servoing? Why not use force/torque measurements? How is the simplified action space mapped back to the robot 14-DoF joint space and how are the desired end-effector commands controlled/tracked?


**Summary Of Recommendation:**

This paper presents a novel approach to scooping food items with a bimanual arm setting including an active pusher and a scooper arm. The approach is capable of handling a variety of food items as demonstrated in the videos and shows improved performance over single-arm scooping and baselines. However, some issues need to be addressed regarding clarification of the framework, dataset generation and learning.

---

> ### Author Response · Authors · 2022-08-22
> **Response to Reviewer FpM9**
>
> We thank the reviewer for their time and detailed feedback. We appreciate the reviewer’s recognition of the variety of complexities and failures of food manipulation, including breakage, falling out of a utensil, and rolling out of the workspace. We provide the following responses to clarify uncertainties and discuss their ideas. We have incorporated their feedback and these clarifications into the paper in blue to make the writing more clear and comprehensive.
>
> > It is not entirely clear to me what is being learned in the MDP and how that data is acquired. In section 3 the MDP is introduced and states that the transition dynamics are unknown, if they are unknown then they must be learned or parametrized somehow?
>
> We use the MDP as a general model for the problem of bimanual scooping, but never explicitly learn or parametrize the dynamics of the system, i.e., we do not solve for a policy for this MDP. Due to the wide variability of food object geometries and physical properties, food manipulation tasks have complex dynamics. Given that we do not have access to a model for these underlying dynamics, we instead design our policy to be an adaptive bimanual scooping primitive composed of three phases that are defined to be performed in succession. We apologize that the connection between the policy and model was not clear, and we have now added more explicit connections between the MDP definition and our proposed method in the text.
>
> We agree that learning the dynamics of a food manipulation setting could be beneficial for planning purposes, especially with longer horizon tasks like grouping foods on a plate before acquiring. We leave this to future work and include this in our Limitations section.
>
> > Also, first it is mentioned that the action space is 14-DoF but then it is mentioned that it is simplified by the bimanual primitive which has 3 phases, so is the new action space 3 dimensions?
>
> We are controlling two 7-DoF robots, so our robot controller is performing 14-DoF actions. However, the primitive we propose performs three phases in succession, with one of the phases being parametrized by only three variables, $(x_f, y_f)$ and $\alpha$, which represents the food position and the distance traveled by the pusher during the Pushing phase respectively. The scooping primitive performs hand-designed actions relative to the initial food position $(x_f, y_f)$. The $\alpha$ parameter prevents breakage caused by the pusher and scooper tool squeezing a food object between each other during scooping. We find that other common failure modes during scooping can be mitigated with stabilizing strategies, and do not require an additional variable in the parametrization of the scooping primitive. As a result, our policy has a 3 dimensional action space, which is an initial food position $(x_f, y_f)$ and a tuned $\alpha$ parameter. We have added this clarification to the main text in Section 3.
>
> > It is unclear to me if this sequence or transitions between the phases are learned from the food interactions or if they are predefined and parametrized by the designers. How is the MDP deciding when to transition between phases?
>
> We apologize if this wasn’t clear. The transitions between the phases are predefined as part of the bimanual scooping primitive, and are not learned. We have added this clarification to the new paper version in Section 3.
>
> > Related to the previous point, the paper only gives details about the risk r:R..→ {0,1} and failure classifiers  f:R..→{0,1} which both seem to map to a binary classification variable. It is not clear to me how the outputs of these functions are used to modulate α which seems to be a continuous function based on the figures and videos.
>
> We thank the reviewer for pointing out this point of confusion. $\alpha$ is a continuous scaling parameter for the distance traveled by the pusher during the Pushing phase and is adjusted *online* with the Failure Classifier by servoing for a failure-imminent state. Before each scooping action, $\alpha$ is initialized to 1, indicating that the pusher travels the full distance to the spoon mouth. However, when the classifier detects a breakage-imminent state, the Pushing phase is terminated early so the pusher does not travel the full distance. In an early-terminated scooping rollout, $\alpha < 1$ because the pusher did not travel the full distance. For example, if a breakage-imminent state was detected after 65% of the Pushing phase was completed, the pusher tool would have traveled 65% of the full pushing distance, and $\alpha = 0.65$. We have added a more detailed explanation of $\alpha$ in Section 3 of the paper on Lines 217-229.

---

> ### Author Response · Authors · 2022-08-22
> **Response to Reviewer FpM9 (Part 2)**
>
> (continuing response due to character limit, part 2)
>
> > It would be ideal if the authors included a baseline comparison to the work cited from [23] which learns the bimanual interaction constraints from human demonstrations.
>
> We thank the reviewer for the suggestion. We believe the reviewer is referring to [24] instead of [23], but please clarify if that is not the case. We would like to note that comparison would not really be valid due to different assumptions between the methods. In our work, we consider the setting of scooping a variety of foods autonomously and do not assume access to extra human demonstrations. However, [24] is developing their policies based on collected human demonstrations, which we do not have access to. We note that learning the bimanual interaction constraints from these demonstrations is also extremely nuanced and [24] uses specialized sensorized gloves to record these human demonstrations.
>
> > Is there a reason why adaptation to avoid breakage is only considered in the cupping phase?
>
> We find that breakage failures predominantly occur during the Pushing phase due to the Cupping stabilizing strategy. We hypothesize this is because in the Pushing phase, all the objects in the scene (pusher, food item, and scooper) are in contact and breakage is possible. We did not observe breakage failures in other phases during the scooping rollouts.
>
> > It would be interesting to see how the primitive performs when trying to scoop items that are slightly larger than the scoop itself, like 2 strawberries or a larger piece of jello or cheesecake…
>
> Thank you for this suggestion, we agree this would be an interesting case to consider. We ran experiments on scooping 2 strawberries simultaneously, which is larger than our scoop bowl, and have attached the videos to our website at https://sites.google.com/view/bimanualscoop-corl22/home. We find that our system is able to achieve 80% success over 5 rollouts of scooping two strawberries simultaneously.
>
> > it would be great if the authors discussed how the presented strategy can be transferred to other types of tasks or if it can be adapted to handle other types of items that are not food but need a scooping mechanism like soil, sand, etc.
>
> Thank you for this application idea! While we have not considered different objects beyond food yet, we were inspired by the reviewer’s suggestion to try CARBS on foods of different granularities. We ran new experiments on scooping pre-grouped rice and couscous, which have similar properties to sand as they are large groups of multiple small objects. We have added these new experiments to the Appendix (see Appendix G) and have attached the video results from these experiments to our website (https://sites.google.com/view/bimanualscoop-corl22/home). We find that CARBS is able to achieve a 79.0% success rate of scooping rice and couscous (measured by weight), which is comparable to a human single-scoop baseline (85.0% success).
>
> > is the control done purely in position mode with visual servoing?
>
> The control for all parts of the bimanual scooping primitive other than the pushing distance parametrized by $\alpha$ is predefined by the primitive in position mode. $\alpha$ is controlled with visual servoing.
>
> > Why not use force/torque measurements?
>
> We report additional force/torque readings during scooping rollouts of 2 robust food classes (grape and strawberry) and fragile tofu. In Figure 10, we find that the force-torque readings are too noisy to be used to replace the Risk Classifier and distinguish between robust and fragile food items. In Figure 11, we find that even within fragile food scooping rollouts, we are unable to identify breakage failures from force-torque sensing alone, suggesting the need for a vision-based system for our Failure Classifier. We believe this is due to multi-robot-object interaction nature of bimanual scooping, and such information might be more useful in other types of food acquisition, e.g., skewering as in [1]. We agree multimodal sensing is a promising direction for food acquisition, especially for generalizing to unseen food items, and have highlighted this in our Limitations section.
>
> > How is the simplified action space mapped back to the robot 14-DoF joint space and how are the desired end-effector commands controlled/tracked?
>
> We use an impedance controller to perform our parametrized scooping primitive on two Panda arms. We have added this detail to our implementation section on Line 256.
>
> We thank the reviewer for their thoughtful feedback and suggestions, which we have incorporated to improve the clarity of the attached, new paper version.
>
> [1] E. Gordon, S. Roychowdhury, T. Bhattacharjee, K. Jamieson, and S. S. Srinivasa. “Leveraging Post Hoc Context for Faster Learning in Bandit Settings with Applications in Robot-Assisted Feeding.” ICRA 2021.

---

> ### Author Response · Authors · 2022-08-25
> **Follow Up to Response to Reviewer FpM9**
>
> We hope the new scooping experiments (2 strawberries, rice, and couscous) [ [Full Experiments](https://drive.google.com/file/d/1bFUrXvBjUl5USf_6wapXynmbyYNpaQZc/view?usp=sharing) ] and methodology and implementation clarifications in our updated paper version have addressed your concerns. Because there is only one day left in the rebuttal period, we are following up to see if you have any additional feedback or suggestions we can discuss to improve our paper.

---

### Meta-Review · Area_Chair_W4b5 · 2022-08-04

**Recommendation:** Accept (Poster)
**Confidence:** 5

**Metareview:**

Strengths:

* bimanual solution to food acquisition (new)
* balancing manual design, a lot of engineering, simple learning: it's effective
* good experimental results and tests with real food

Weaknesses:

* a lack of specific explanation of the methodology (e.g., how is a primitive described/implemented? how is theta set?)
* lacking detail on the robot control (especially, where the end-effector control point is set, if it is torque controlled, impedance controlled etc.)
* lacking detail on the framework, dataset generation and learning

Summary after rebuttal:
We thank the authors for the updates and the rebuttal. The reviewers and the AC agree that this paper provides an interesting contribution to robotics, balancing engineering and learning. Concerns over generalization remain, so it is recommended as poster.


**Best Paper Nomination:**

No

---

> ### Author Response · Authors · 2022-08-22
> **Response to Area Chair W4b5**
>
> We thank everyone for their reviews. We are excited that the reviewers agree that our scooping system is an effective solution to a challenging task with convincing results. While we have prepared individual responses for each of the reviewers, we address common concerns here.
>
> *Methods and Framework Explanation Concerns (Reviewer FpM9, aovx, rkd9)*: Several reviewers have points of confusion on the stabilizing strategies and $\alpha$ parameter of the bimanual scooping primitive. We have added more detail on this in the main text and have updated Figure 1 to more clearly visualize the data flow of the system at runtime. We also added new experiments ablating the stabilizing strategies to the Appendix to more clearly motivate what types of failures each strategy is aimed at preventing.
>
> *Lack of Implementation Details (Reviewer FpM9, 1gVW, rkd9)*: We have updated the main text of the paper in blue with additional implementation details describing robot control and the theta value of our Angled Pusher strategy. We have added a new figure to the Appendix describing the data augmentation techniques used to create our datasets, along with examples of augmented images.
>
> *Multimodal Sensing (Reviewer FpM9, rkd9)*: Multiple reviewers suggested incorporating tactile feedback to this system to sense breakage or food fragility. We present new plots of force readings in the Appendix. We have previously looked into using this data as well. However, we show in these plots that these readings are too noisy (due to close mutli-robot-object interactions) to replace vision-based methods for scooping, and we highlight this in the Limitations section.
>
> We have additionally incorporated all other reviewer comments in the draft in blue, and have added new experiments in the Appendix. Thank you!